# The Functional Role of the Long Non-Coding RNA LINCMD1 in Leiomyoma Pathogenesis

**DOI:** 10.3390/ijms252111539

**Published:** 2024-10-27

**Authors:** Tsai-Der Chuang, Nhu Ton, Shawn Rysling, Omid Khorram

**Affiliations:** 1The Lundquist Institute for Biomedical Innovation, Torrance, CA 90502, USA; tchuang@lundquist.org (T.-D.C.); nhu.ton@lundquist.org (N.T.); shawn.rysling@lundquist.org (S.R.); 2Department of Obstetrics and Gynecology, David Geffen School of Medicine at University of California, Los Angeles, CA 90095, USA

**Keywords:** fibroid, LINCMD1, miR-135, APC, Wnt/β-catenin signaling, ECM

## Abstract

Existing evidence indicates that LINCMD1 regulates muscle differentiation-related gene expression in skeletal muscle by acting as a miRNA sponge, though its role in leiomyoma development is still unknown. This study investigated LINCMD1′s involvement in leiomyoma by analyzing paired myometrium and leiomyoma tissue samples (n = 34) from patients who had not received hormonal treatments for at least three months prior to surgery. Myometrium smooth muscle cells (MSMCs) were isolated, and gene expression of LINCMD1 and miR-135b was assessed via qRT-PCR, while luciferase assays determined the interaction between LINCMD1 and miR-135b. To examine the effects of LINCMD1 knockdown, siRNA transfection was applied to a 3D MSMC spheroid culture, followed by qRT-PCR and Western blot analyses of miR-135b, APC, β-Catenin and COL1A1 expression. The results showed that leiomyoma tissues had significantly reduced LINCMD1 mRNA levels, regardless of patient race or MED12 mutation status, while miR-135b levels were elevated compared to matched myometrium samples. Luciferase assays confirmed LINCMD1′s role as a sponge for miR-135b. LINCMD1 knockdown in MSMC spheroids increased miR-135b levels, reduced APC expression, and led to β-Catenin accumulation and higher COL1A1 expression. These findings highlight LINCMD1 as a potential therapeutic target to modulate aberrant Wnt/β-Catenin signaling in leiomyoma.

## 1. Introduction

Leiomyomas, also known as fibroids, are benign uterine tumors affecting approximately 70% of women during their reproductive years. There are currently no effective long-term medical therapies available and hysterectomy is the only definitive treatment available [1,2]. The etiology of leiomyomas remains unknown, but they are known to have a higher prevalence and symptom severity in African American women. These tumors depend on ovarian steroids for growth and exhibit fibrotic characteristics, with excess cell proliferation, inflammation, angiogenesis, and extracellular matrix (ECM) accumulation playing central roles in their pathogenesis [1,2,3].

The mechanisms underlying the aberrant expression of protein-coding genes that regulate these processes in leiomyomas have been the focus of intense research. Moreover, genomic studies have identified chromosomal rearrangements and mutations associated with leiomyoma development and progression, including mutations in *MED12* (Mediator Complex Subunit 12), *HMGA2* (High-Mobility Group AT-Hook 2), *FH* (Fumarate Hydratase), and *COL4A5/6* (Collagen Type IV Alpha 5 and Alpha 6) [4,5,6]. *MED12* mutations, which occur in up to 80% of leiomyomas, are associated with abnormal activation of Wnt/β-Catenin signaling, sex steroid receptor signaling, and increased expression of genes related to cell proliferation and fibrosis [4,7,8].

In addition to protein-coding genes, several studies have revealed the misexpression of non-coding RNAs (ncRNAs) in leiomyomas [9,10,11]. Thousands of ncRNAs have been identified in the human genome, including small ncRNAs (sncRNAs) and long ncRNAs (lncRNAs) [12,13,14,15]. LncRNAs, which are transcripts longer than 200 nucleotides that are transcribed from various genomic regions and can overlap with or be intergenic, intronic, or antisense to protein-coding genes [12,13]. The evidence suggests that both sncRNAs, particularly miRNAs, and lncRNAs regulate protein-coding genes through transcriptional, posttranscriptional, and epigenetic mechanisms [12,13]. LncRNAs can also act as miRNA sponges or competing endogenous RNAs (ceRNAs), preventing miRNAs from binding to their target mRNAs [16,17,18]. These regulatory activities occur in a cell- and tissue-specific manner and are involved in various physiological processes, while their dysregulation has been linked to tumorigenesis and tissue fibrosis [16,17,18].

The functional role of lncRNAs in leiomyoma pathogenesis is an emerging area of research. Cao et al. reported that the lncRNA *H19* is upregulated in leiomyomas and is associated with an increased expression of HMGA2 and other genes related to proliferation, inflammation, and ECM deposition via TET3-mediated epigenetic modifications [19]. Our previous studies identified a profibrotic network involving *MIAT*, miR-29, TGF-β3, and collagens in leiomyomas, which forms a positive feedback loop that causes continuous fibrosis and the growth of leiomyomas [20]. Additionally, we reported that another lncRNA *XIST* (X-inactive Specific Transcript) which is overexpressed in leiomyomas acts as a sponge for miR-29c and miR-200c. This interaction leads to an increased expression of their targets, including COL1A1, COL3A1, and FN1 (Fibronectin) [21]. Using next-generation RNA sequencing, we developed a comprehensive profile of differentially expressed sncRNAs and lncRNAs in leiomyomas, demonstrating the impact of race/ethnicity and *MED12* mutation on their expression and highlighting their significance in leiomyoma pathogenesis [10,11,22]. The dysregulated expression of lncRNAs in leiomyomas through interactions with miRNAs, mRNAs, DNA, and proteins may affect the expression of protein-coding genes linked to cell proliferation, extracellular matrix accumulation, and inflammation in leiomyomas. Our overarching goal is to identify lncRNA-miRNA networks that regulate protein-coding genes which have a physiological significance in leiomyoma pathogenesis, particularly those involved in ECM composition, cell proliferation, and inflammation.

Aberrant activation of Wnt/β-Catenin signaling is well documented in leiomyomas [23,24]. Understanding the regulatory role of ncRNAs in Wnt/β-Catenin signaling in leiomyomas is critical but currently lacking. In this study, we hypothesized that leiomyomas express lower levels of LINCMD1 compared to the myometrium and that LINCMD1, through a miR-135-guided mechanism, regulates the expression of genes associated with Wnt/β-Catenin signaling activation. To test this hypothesis, we analyzed a large cohort of paired leiomyoma specimens and performed functional testing using a three-dimensional (3D) culture system, as characterized by Vidimar et al. [25], which better represents the in vivo state by maintaining physiological cell-to-cell interactions and gradients of oxygen, nutrients, and catabolites [26,27].

## 2. Results

### 2.1. An Inverse Expression Relationship Between LINCMD1 and miR-135b in Leiomyomas

A previous study suggested that LINCMD1 functions as a sponge for miR-135b, playing a role in skeletal muscle differentiation [28]. To explore the relationship between LINCMD1 and these miRNAs in leiomyomas, we first assessed their expression levels. Using qRT-PCR, we confirmed that LINCMD1 is expressed in both leiomyomas and matched myometrium, but at significantly lower levels in leiomyomas (Figure 1A). Conversely, miR-135b (Figure 1B), was significantly upregulated in leiomyomas compared to the matched myometrium. The differential expression of LINCMD1 was independent of race/ethnicity and the MED12 mutation status of the leiomyomas. Next, we investigated the functional relationship between LINCMD1 and miR-135b in leiomyomas, given their inverse expression pattern. To explore this, we knocked down LINCMD1 in myometrial spheroid cells using LINCMD1 siRNA (Figure 2A). The transfection of siRNA targeting LINCMD1 decreased the expression of LINCMD1 by 62.5%. Following LINCMD1 knockdown, there was a significant increase in the expression of miR-135b (Figure 2B).

### 2.2. LINCMD1 Directly Targets miR-135b and Regulates Wnt/β-Catenin Signaling in Leiomyomas

A bioinformatic analysis for lncRNA-miRNA interactions indicated complementary base pairing between LINCMD1 and miR-135b (Figure 3A) [29,30]. To experimentally confirm this interaction, we used the luciferase reporter assay, which demonstrated that LINCMD1 directly targets miR-135b, confirming its sponge effect (Figure 3B).

Previous studies have reported that the *APC* gene is a target of the miR-135 family in other cell types and plays a role in regulating Wnt/β-Catenin signaling, which is known to be dysregulated in leiomyomas [23,24,31,32,33]. However, the function of miR-135b in leiomyomas had not been previously explored. To investigate this, we overexpressed miR-135b by transfecting myometrial spheroid cells with pre-miR-135b oligonucleotides and assessed its effects on the expression of APC and Wnt/β-Catenin signaling-related proteins (Figure 3C,D). Our results showed that the overexpression of miR-135b resulted in a significant reduction in APC expression, along with an activation of Wnt/β-Catenin signaling in myometrial spheroid cells (Figure 3C,D). We also examined the expression of APC and β-Catenin in paired myometrium and leiomyoma tissues and found that APC mRNA expression was reduced, while β-Catenin levels were upregulated in leiomyomas compared to matched myometrium. This was independent of race/ethnicity and MED12 mutation status (Figure 4A,B). Additionally, APC protein levels were decreased in leiomyomas, while levels of non-phosphorylated β-Catenin at Ser45 (active form), total β-Catenin, and COL1A1 were increased (Figure 4C,D). Non-phosphorylated β-Catenin at Ser45 is resistant to proteasomal degradation and remains functionally active in the Wnt/β-Catenin signaling pathway [34,35]. COL1A1 was measured due to its role in ECM deposition and as a direct downstream target of Wnt/β-Catenin signaling [36,37,38].

Knockdown of LINCMD1 in myometrial spheroid cells resulted in decreased APC expression and increased levels of total β-Catenin, non-phosphorylated β-Catenin at Ser45 (activated), and COL1A1 (Figure 5A–C). To determine if LINCMD1 regulates Wnt/β-Catenin signaling in leiomyomas through miR-135b, we performed LINCMD1 knockdown in primary MSMC spheroid cells and subsequently transfected these cells with either anti-miR-135b or scrambled oligonucleotides as a control. As shown in Figure 5C,D, the anti-miR-135b transfection following LINCMD1 knockdown partially reversed the protein levels of APC, non-phosphorylated β-Catenin at Ser45, total β-Catenin, and COL1A1, supporting the existence of a LINCMD1/miR-135/APC/β-Catenin signaling axis in leiomyomas which contributes to ECM accumulation in leiomyomas.

## 3. Discussion

Our data indicate that LINCMD1 functions as a molecular sponge for the miR-135b in leiomyomas. The decreased expression of LINCMD1 in leiomyoma tissue leads to an increase in miR-135b levels, which subsequently downregulates its target gene, APC. APC is a critical negative regulator of the Wnt/β-Catenin signaling pathway through its role in promoting β-Catenin degradation. When APC expression is reduced, the degradation of β-Catenin is impaired, resulting in the activation of Wnt/β-Catenin signaling. This pathway is a key driver in the pathogenesis of leiomyoma and contributes to the abnormal accumulation of ECM, a hallmark of leiomyoma development.

LINCMD1 is expressed in the cytoplasm of differentiating myogenic cells and newly regenerated myofibers during myoblast differentiation, where it functions as a competitive endogenous RNA for miR-133b and miR-135b [28]. These microRNAs target two key genes involved in myogenic differentiation: *MAML1* (mastermind-like protein 1) and *MEF2C* (myocyte enhancer factor 2C) [28]. Additionally, research has shown that LINCMD1 is protected from DROSHA-mediated cleavage when bound to the HuR protein. However, as myoblast differentiation advances, upregulation of miR-133 inhibits HuR transcription, leading to the downregulation of LINCMD1 [39]. This interaction plays a critical role in regulating the progression of myogenic differentiation from early to more advanced stages. While the LINCMD1 sequence in mice contains two binding sites for miR-135b and one for miR-133b [28], no predicted complementary binding site for miR-133b is found in the human LINCMD1 sequence. This absence may explain our findings, where an inverse correlation between LINCMD1 and miR-133b expression levels was not detected in leiomyomas as compared to matched myometrium.

In addition to LINCMD1, several lncRNAs have been identified as modulators of the Wnt/β-Catenin signaling pathway, which could contribute to cancer prognosis and diagnosis [40]. For instance, the altered expression of *LINC01133* in gastric cancer regulates the Wnt/β-Catenin signaling pathway through the miR-106a-3p/APC axis [41]. Similarly, *HOTAIR* was recently reported to modulate the Wnt/β-Catenin signaling pathways in gastric cancer by sponging miR-34a [42]. *SNHG1* promotes non-small cell lung cancer tumorigenesis and progression via the miR-101-3p/SOX9/Wnt/β-Catenin axis [43]. *CASC15* has been observed to regulate the expression of leucine-rich repeat-containing G-protein coupled receptor 5 (LGR5) by targeting miR-4310, thereby activating the Wnt/β-Catenin signaling pathway in colon cancer [44]. Furthermore, *CASC15* correlates with β-Catenin expression in melanoma, promoting melanoma progression through activation of the Wnt/β-Catenin signaling pathway [45]. Our own studies have shown that *CASC15* is upregulated in leiomyomas [46]. In cervical cancer, the lncRNA *NEAT1* has been identified as a contributor to tumor progression by activating the WNT/β-Catenin/PDK1 signaling axis [47]. In leiomyomas, the lncRNA *APTR* is overexpressed and enhances cell proliferation by activating the Wnt/β-Catenin pathway through targeting estrogen receptor alpha (ERα) [48]. Additionally, the expression level of the lncRNA *SRA1* has been observed to correlate with the *MED12* mutation status in leiomyomas [49].

The miR-135 family has been shown to indirectly influence the activity of the Wnt/β-Catenin signaling pathway by targeting APC, thereby promoting cancer progression [34,35]. APC functions as a key regulator of the Wnt/β-Catenin signaling pathway. Mutations in the *APC* gene result in aberrant β-Catenin activity, driving the development of cancer and other human diseases [50]. Research suggests that the expression of miRNAs and their regulatory roles are highly tissue-specific and context-dependent, highlighting the complexity of their functions across different cellular environments [51,52]. One study found that circulating miR-135a levels were significantly reduced in the serum of women with endometriosis compared to controls [53]. Our current study is the first to demonstrate that leiomyomas exhibit increased levels of miR-135b, which is involved in the activation of the Wnt/β-Catenin signaling pathway [23,24]. In line with our findings, β-Catenin expression was significantly increased, while APC expression was decreased in leiomyomas, with no correlation with MED12 mutation status [54]. Additionally, we observed a correlation between elevated COL1A1 expression and decreased LINCMD1 levels in leiomyomas, which could be reversed by correcting miR-135b levels (Figure 5). Notably, the Wnt-β-Catenin signaling pathway has been reported to regulate COL1A1 expression [55]. Myocardin-related transcription factor A (MRTF-A), the co-activator of serum response factor (SRF), has been shown to be induced by the Wnt-β-Catenin signaling pathways and to promote COL1A1 expression by directly regulating chromatin acetylation and recruiting RNA polymerase II to the COL1A1 promoter in human breast cancer cells [55]. Another study showed that inhibiting Wnt/β-Catenin signaling through Wnt inhibitor treatment decreased COL1A1 expression in synovial fibroblasts isolated from osteoarthritis (OA) patients [56].

In conclusion, our study uncovers a novel regulatory axis involving the long non-coding RNA LINCMD1 and miR-135b, which modulates the Wnt/β-Catenin signaling pathway and contributes to ECM deposition in leiomyomas. This mechanistic insight provides a promising foundation for targeting non-coding RNAs, such as LINCMD1, to correct the abnormal activation of the Wnt/β-Catenin pathway, and as a potential therapeutic target for leiomyomas.

## 4. Materials and Methods

### 4.1. Tissue Collection

Uterine leiomyoma and paired myometrium samples were collected from 34 premenopausal patients at Harbor-UCLA Medical Center, none of whom had been on hormonal medications for at least three months prior to surgery. Institutional Review Board approval (#036247) was obtained from the Lundquist Institute at Harbor-UCLA Medical Center, and informed consent was secured from all participants before surgery. The clinical data of all patients enrolled in this study including age, racial/ethnic background, and MED12 mutation status is shown in Appendix A. The tissue samples were either snap-frozen and stored in liquid nitrogen for later analysis or used to isolate MSMC, as described previously [57,58].

### 4.2. Reagents and Spheroid Cell Culture

MSMCs were cultured in DMEM supplemented with 10% fetal bovine serum, with media changes every 2–3 days, until confluence was reached. Cells at passages p1 to p3 were used for all experiments. The isolated MSMCs were seeded into 6-well plates (1.5 × 10^5^ cells/well) coated with 0.5% agarose gel and incubated for 48 h to facilitate spheroid formation, with spheroids ranging in size from 50 µm to 250 µm in diameter [21]. Each cell culture experiment was performed at least three times using three different MSMC isolated from separate patients. All reagents and supplies for isolation and cell culture were obtained from Sigma-Aldrich (St. Louis, MO, USA), Invitrogen (Carlsbad, CA, USA), and Fisher Scientific (Atlanta, GA, USA).

### 4.3. siRNA Transfection

Before spheroid formation, primary MSMCs were transfected with 50 nM of either siRNA negative control (siNC) or siRNA targeting LINCMD1 (siLINCMD1; 5′-GAGCUGACUAAGAAGAAGAAACUCC-3′) using PureFection transfection reagent (System Biosciences, Inc., Mountain View, CA, USA), following the manufacturer’s instructions [20].

### 4.4. RNA Isolation and qPCR Analysis

Total RNA was extracted from leiomyomas and matched myometrium tissues using Trizol (Thermo Fisher Scientific, Waltham, MA, USA). RNA concentration and integrity were assessed using a Nanodrop 2000c spectrophotometer (Thermo Scientific) and an Agilent 2100 Bioanalyzer (Agilent Technologies, Santa Clara, CA, USA), following the manufacturer’s protocols as previously described [11]. Subsequently, 1 μg of RNA was reverse transcribed using random primers for LINCMD1, Adenomatous Polyposis Coli (APC), and COL1A1. Primer design for miR-135 and the PCR conditions have been described previously [59]. Quantitative PCR was performed using SYBR gene expression master mixes (Applied Biosystems, Carlsbad, CA, USA). The reactions were incubated at 95 °C for 10 min, followed by 40 cycles of 15 s at 95 °C and 1 min at 60 °C. mRNA and miRNA levels were quantified using the Applied Biosystems 7500 Fast Real-Time PCR System, and normalized to FBXW2 [60] for mRNA and RNU6B [20] for miRNA. All reactions were run in triplicate, and relative expression levels were calculated using the comparative cycle threshold method (2^−ΔΔCq^), as recommended by Applied Biosystems. Expression values were reported as fold changes compared to the corresponding control group. The primer sequences used in this study were designed by PrimerQuest Tool (Integrated DNA Technologies, Inc. Coralville, IA, USA) as shown in Appendix A.

### 4.5. Immunoblotting

Total protein isolated from MSMC spheroids was subjected to immunoblotting as previously described [61]. Briefly, the samples were resuspended in RIPA buffer containing 1 mM EDTA and EGTA (Boston BioProducts, Ashland, MA, USA), supplemented with 1 mM PMSF and a complete protease inhibitor cocktail (Roche Diagnostics, Indianapolis, IN, USA), sonicated, and centrifuged at 14,000 rpm for 10 min at 4 °C. Protein concentration was measured using the BCA™ Protein Assay Kit (Thermo Scientific Pierce, Rockford, IL, USA). Equal amounts of total protein (30 µg per sample) were denatured with SDS-PAGE sample buffer and separated by electrophoresis on an SDS polyacrylamide gel. After transferring the proteins to a nitrocellulose membrane, the membrane was blocked with TBS-Tween containing 5% milk and incubated with the following primary antibodies: APC (1:3000, Proteintech Group, Inc., Chicago, IL, USA), non-phosphorylated β-Catenin at Ser45 (1:1500, Cell Signaling Technology, Danvers, MA, USA), total β-Catenin (1:3000, Cell Signaling Technology, Danvers, MA, USA), and COL1A1 (1:3000, Proteintech Group, Inc., Chicago, IL, USA). After each antibody incubation, the membranes were washed with TBS containing 0.1% Tween-20. Detection was performed using SuperSignal West Pico Chemiluminescent Substrate™ (Thermo Scientific Pierce), and protein bands were visualized via photographic emulsion and quantified by densitometry. The membranes were also stripped and probed with an anti-GAPDH antibody (1:3000) as a loading control. Specific protein band densities were measured using the ImageJ software version 1.54k (http://imagej.nih.gov/ij/) (accessed on 3 October 2024) and normalized to GAPDH or Ponceau S staining. Data are presented as means ± SEM, with values expressed as ratios relative to the control group, set at 1.

### 4.6. Reporter Plasmid Construction

The recombinant luciferase reporter plasmid pEZX-MT01 (LINCMD1) was generated by inserting an EcoRI/XhoI-digested, PCR-amplified fragment of LINCMD1 (ENSG00000225613), which includes the miR-135b binding site, downstream of the luciferase reporter in the pEZX-MT01 vector (GeneCopoeia, Rockville, MD, USA).

### 4.7. Luciferase Reporter Assays

Before spheroid formation, primary MSMC were transfected with 50 nM of 2′-O-methoxyethyl modified pre-miR-135b oligonucleotides or the corresponding pre-miR negative control (NC) (Applied Biosystems, Carlsbad, CA, USA) using PureFection transfection reagent, as previously described [62]. Simultaneously, the cells were co-transfected with a luciferase reporter plasmid (1 μg/well), either pEZX-MT01 (Control) or pEZX-MT01 (LINCMD1). After 48 h, Firefly and Renilla luciferase activities were measured using the Dual-Luciferase Reporter Assay System (Promega, Madison, WI, USA). Firefly luciferase activity was normalized to Renilla luciferase activity, and the results were expressed as the mean ± SEM from three independent experiments, each performed in triplicate using cells isolated from different patients. The induction level was compared to the ratio in cells transfected with the negative control, which was independently set to 1.

### 4.8. Statistics and Power Analysis

All data throughout the text are presented as mean ± SEM and were analyzed using GraphPad Prism 10 software (GraphPad, San Diego, CA, USA). The normality of the datasets was assessed using the Kolmogorov–Smirnov test. Since the data in Figure 1 and Figure 4 were not normally distributed, non-parametric tests were employed for analysis. Comparisons between two groups were performed using the Wilcoxon matched pairs signed rank test. For normally distributed data (Figure 2, Figure 3 and Figure 5), comparisons between two groups were analyzed using unpaired Student’s *t*-tests, and one-way ANOVA was used for comparisons involving multiple groups. Statistical significance was defined as *p* < 0.05. Assuming a minimal detectable difference of 25% between leiomyoma and myometrium pairs, with an expected standard deviation of 25%, at least 16 paired samples are required to achieve a statistical power of 0.80 at a significance level of 0.05 using a two-sided *t*-test.

## Figures and Tables

**Figure 1 ijms-25-11539-f001:**
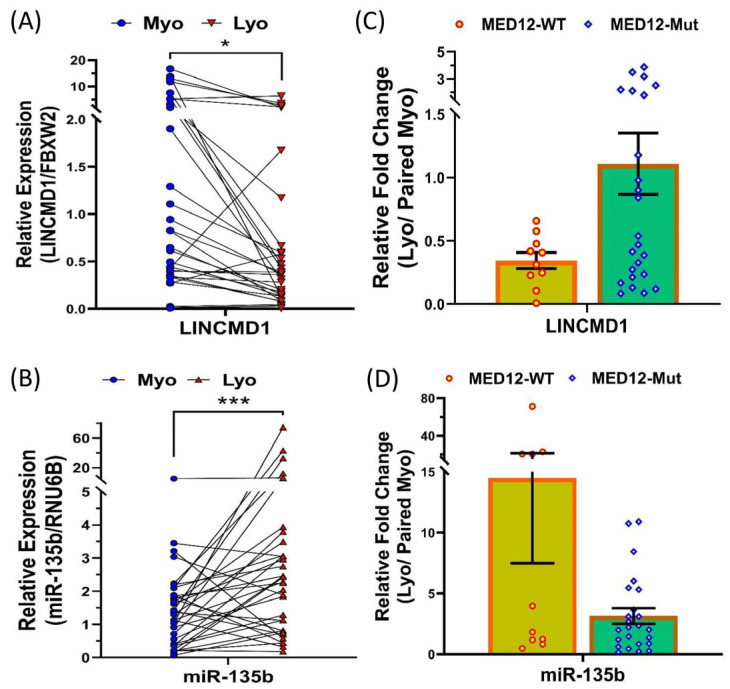
An inverse relationship in the expression of LINCMD1 and miR-135b in leiomyomas. qRT-PCR analysis showing the expression levels of lncRNA LINCMD1 (**A**) and miR-135b (**B**) in 34 paired samples of myometrium (Myo) and leiomyomas (Lyo). Relative fold change in LINCMD1 (**C**) and miR-135b (**D**) expression (Lyo/paired Myo) is displayed based on MED12 mutation status, comparing MED12 wild type (n = 10) and MED12 mutated (n = 24) samples. Data are presented as mean ± SEM, with statistical significance indicated as * *p* < 0.05 and *** *p* < 0.01.

**Figure 2 ijms-25-11539-f002:**
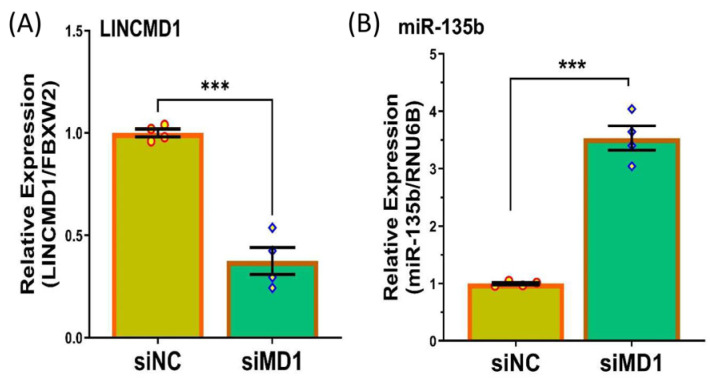
siRNA-mediated knockdown of LINCMD1 in myometrium spheroid cells for 96 h significantly reduced LINCMD1 expression levels (**A**) and led to an increased expression of miR-135b (**B**). Data (n = 4) are presented as mean ± SEM, with *** *p* < 0.01 indicating statistical significance.

**Figure 3 ijms-25-11539-f003:**
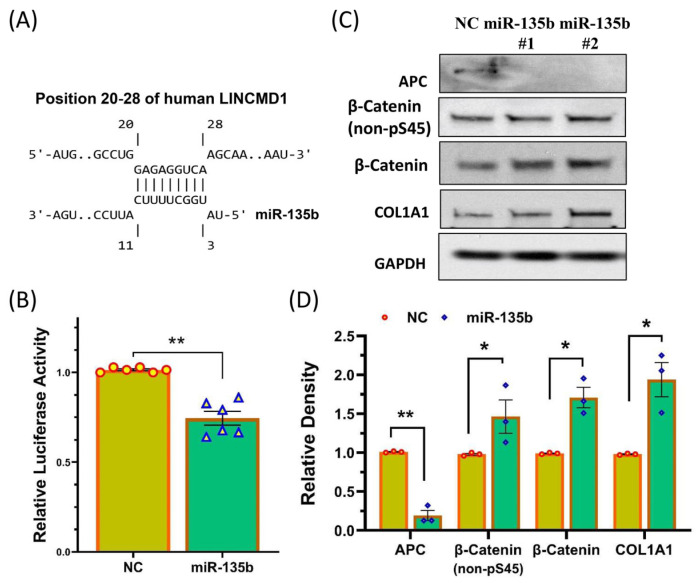
LINCMD1 directly targets miR-135b, and the resulting changes in miR-135b regulate the β-Catenin signaling pathway. (**A**) Sequence alignment showing the coordinated positions of LINCMD1 with miR-135b. (**B**) Relative luciferase activity in myometrium spheroid cells transfected with Renilla and Firefly luciferase reporter pEZX-MT01 (Control) or pEZX-MT01 (LINCMD1). The ratio of Firefly to Renilla luciferase activity was measured after 48 h and expressed as relative luciferase activity compared to NC, which was independently set to 1 (n = 6). (**C**) Western blot analysis of APC, non-phosphorylated β-Catenin at serine 45, total β-Catenin, and COL1A1 following the transfection of myometrium spheroid cells with control pre-miR oligonucleotides (NC) or pre-miR-135b for 96 h. The relative band intensities are presented in a bar graph (**D**, n = 3) as mean ± SEM. * *p* < 0.05; ** *p* < 0.01.

**Figure 4 ijms-25-11539-f004:**
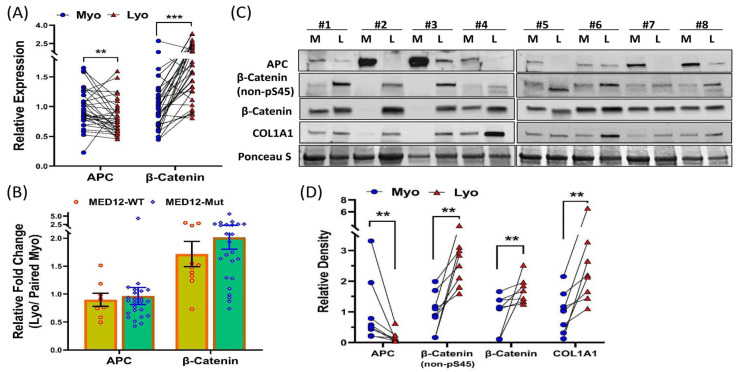
An inverse relationship between APC expression and β-Catenin activity in leiomyomas. (**A**) qRT-PCR analysis showing the expression levels of APC and β-Catenin in 34 paired myometrium (Myo) and leiomyoma (Lyo) samples. ** *p* < 0.01; *** *p* < 0.001. (**B**) Their relative expression is shown as fold change (Lyo/paired Myo) based on MED12 mutation status. (**C**) Western blot analysis of APC, non-phosphorylated β-Catenin at serine 45, total β-Catenin, and COL1A1 in tissue extracts (n = 8) from myometrium (M) and paired leiomyoma (L). The relative band intensities are displayed in (**D**), with data presented as mean ± SEM. ** *p* < 0.01.

**Figure 5 ijms-25-11539-f005:**
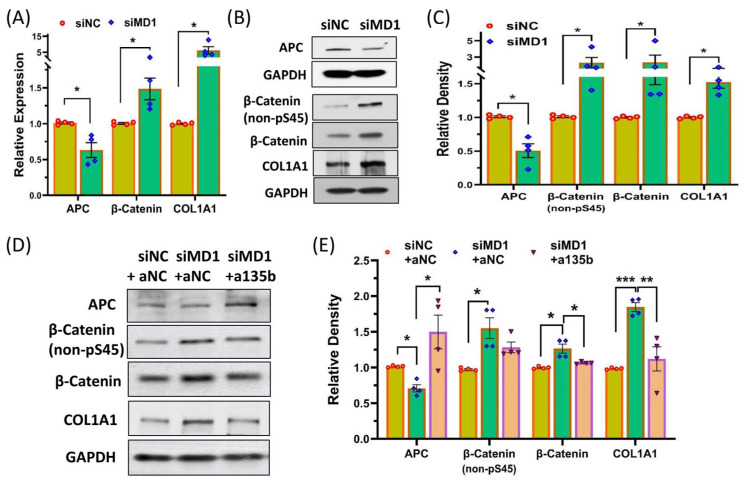
The LINCMD1/miR-135/APC/β-Catenin signaling axis in leiomyomas contributes to ECM accumulation. (**A**) qRT-PCR analysis showing mRNA expression levels of APC, β-Catenin, and COL1A1 in primary myometrium spheroid cells following transfection with LINCMD1 siRNA for 96 h (n = 4, * *p* < 0.05). (**B**) Representative Western blot analysis of β-Catenin, non-phosphorylated β-Catenin at serine 45, total β-Catenin, and COL1A1 after LINCMD1 siRNA transfection in primary myometrium spheroid cells for 96 h. Relative band intensities are shown in (**C**), with data presented as mean ± SEM (n = 4, * *p* < 0.05). (**D**) A representative gel showing levels of APC, non-phosphorylated β-Catenin at serine 45, total β-Catenin, and COL1A1 after co-transfection of LINCMD1 siRNA with either anti-NC or anti-miR-135b in primary myometrium spheroid cells for 96 h. (**E**) Bar plot depicting the mean relative band intensities, presented as mean ± SEM (n = 4, * *p* < 0.05; ** *p* < 0.01; *** *p* < 0.001).

## Data Availability

Raw data were generated at The Lundquist Institute. Derived data supporting the findings of this study are available from the corresponding author O.K on request.

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
