# Peer review of "The Functional Role of the Long Non-Coding RNA LINCMD1 in Leiomyoma Pathogenesis"

_ijms, 2024, doi:10.3390/ijms252111539_

Round 1

Reviewer 1 Report

Comments and Suggestions for Authors

The manuscript shows that:

-Lyo patients have significantly lower LINCMD1 expression than Myo patients and higher miR-135b.

-The MED12 mutation significantly increases LINCMD1 expression and decreases that of miR-135b.

-LINCMD1 and miR-135b are inversely related as shown by siRNA.

-Transfection with LINCMD1 blunts miR-135b expression and APC protein level, but increases β-catenin and COL1A1 levels. 

-These changes in protein levels are then compared between two selected groups of Myo and Lyo patients.

-Silencing LINCMD1 blunts APC and raises β-catenin and COL1A1 expression.

A few suggestions for the authors:

1. I suggest the authors add a table for patient characteristics. It will make it much easier for readers.

2. "4.1. Tissues Collection" - Tissue.

3. "Each cell culture experiment was performed at least three times using MSMC derived from different patients" - in some figures the number of samples is too little (n = 4 in Fig. 2 and Fig. 5; n = 3 in Fig. 3) to draw statistically meaningful conclusions which requires justification in the statistical analysis subsection. 

4. Please explain the distinction between n and N (e.g., Fig. 4) in the Statistical analysis subsection.

5. "Results were obtained from five independent experiments using cells from different patients" - the authors must explicitly state in the manuscript the basis upon which patient samples were selected for analysis (e.g., Western blot). Why these 8 patients (Fig. 4) of all subjects?

6. "4.3. siRNA Transfection" - please provide a reference if the kit has been used before.

7. "and RNU6B for miRNA" - please provide a reference for this housekeeping transcript.

8. Primer sequences are better represented in a table. Also state whether each was either predesigned (mention manufacturer) or designed by the authors (mention details; at least software/tool used).

9. It would be better if the authors actually named their figures (e.g., Figure X. [STATE MAIN FINDING]).

10. If LINCMD1 and miR-135b are inversely related (Figs. 1 and 2), then how do the authors explain the seemingly discrepant findings in Figs. 4 and 5? In Fig. 4, when cells are transfected with LINCMD1 (Fig. 3), miR-135b expression and APC protein levels were decreased but β-catenin and COL1A1 levels were increased. However, in Fig. 5, when LINCMD1 was silenced, APC still decreased while β-catenin and COL1A1 increased at both the transcriptional and translational levels. It's only when anti-miR-135b was used that APC levels made sense. The other two proteins, although decreased, were still apparently not significantly different from siNC (which should have been compared [siNC vs. siMD1+a135b] by the authors). I think  this suggests that LINCMD1 and miR-135b might be related albeit through unknown mediator(s). The authors must either refute this in a response or explain it in the Discussion.

Author Response

Reviewer #1
The manuscript shows that:

-Lyo patients have significantly lower LINCMD1 expression than Myo patients and higher miR-135b.

-The MED12 mutation significantly increases LINCMD1 expression and decreases that of miR-135b.

-LINCMD1 and miR-135b are inversely related as shown by siRNA.

-Transfection with LINCMD1 blunts miR-135b expression and APC protein level, but increases β-catenin and COL1A1 levels.

-These changes in protein levels are then compared between two selected groups of Myo and Lyo patients.

-Silencing LINCMD1 blunts APC and raises β-catenin and COL1A1 expression.

A few suggestions for the authors:

  1. I suggest the authors add a table for patient characteristics. It will make it much easier for readers.

Response: Thanks for the suggestions. The clinical data of all patients enrolled in this study including age, racial/ethnic back-ground, and MED12 mutation status is shown in Supplementary Table S1.

  1. "4.1. Tissues Collection" - Tissue.

Response: Thanks for the suggestions. We have corrected it.

  1. "Each cell culture experiment was performed at least three times using MSMC derived from different patients" - in some figures the number of samples is too little (n = 4 in Fig. 2 and Fig. 5; n = 3 in Fig. 3) to draw statistically meaningful conclusions which requires justification in the statistical analysis subsection.

Response: Thank you for your comments, and we apologize for any confusion caused. The numbers indicated in figure legends represent different MSMC isolated from separate patients used to perform the experiments (biological replicates), rather than technical replicates of MSMC from a single patient in each figure. We have updated the information as below:

“Each cell culture experiment was performed at least three times using three different MSMC isolated from separate patients.”

'Each cell culture experiment was performed at least three times using MSMC from at least three different patients.'

  1. Please explain the distinction between n and N (e.g., Fig. 4) in the Statistical analysis subsection.

Response: It should be n. We have corrected it. Thanks for the suggestions.

  1. "Results were obtained from five independent experiments using cells from different patients" - the authors must explicitly state in the manuscript the basis upon which patient samples were selected for analysis (e.g., Western blot). Why these 8 patients (Fig. 4) of all subjects?

Response: Thank you for your suggestions. We have removed the incorrect sentence. Eight paired tissue specimens were randomly selected to assess protein levels of APC, total β-catenin, β-catenin activity, and COL1A1. In Figure 4, we confirmed an inverse relationship between mRNA expression levels (n=34) and protein expression levels (n=8) of APC and total β-catenin in leiomyomas.

  1. "4.3. siRNA Transfection" - please provide a reference if the kit has been used before.

Response: Thanks for the suggestions. A reference had been added.

  1. "and RNU6B for miRNA" - please provide a reference for this housekeeping transcript.

Response: Thanks for the suggestions. A reference had been added.

  1. Primer sequences are better represented in a table. Also state whether each was either predesigned (mention manufacturer) or designed by the authors (mention details; at least software/tool used).

Response: Thanks for the suggestions. We have updated the information in the M&M section as below:

“The primer sequences used in this study were designed by PrimerQuest Tool (Integrated DNA Technologies, Inc. Coralville, Iowa, USA) as shown in Supplementary Table S2.”

  1. It would be better if the authors actually named their figures (e.g., Figure X. [STATE MAIN FINDING]).

Response: Thanks for the suggestions. We have included a sentence highlighting the main findings in the figure legend, as listed below:

“Figure 1. An inverse relationship in the expression of LINCMD1 and miR-135b in leiomyomas.

Figure 3. LINCMD1 directly targets miR-135b, and the resulting changes in miR-135b regulate the β-catenin signaling pathway.

Figure 4. An inverse relationship between APC expression and β-catenin activity in leiomyomas.

Figure 5. The LINCMD1/miR-135/APC/β-catenin signaling axis in leiomyomas contributes to ECM accumulation.”

  1. If LINCMD1 and miR-135b are inversely related (Figs. 1 and 2), then how do the authors explain the seemingly discrepant findings in Figs. 4 and 5? In Fig. 4, when cells are transfected with LINCMD1 (Fig. 3), miR-135b expression and APC protein levels were decreased but β-catenin and COL1A1 levels were increased. However, in Fig. 5, when LINCMD1 was silenced, APC still decreased while β-catenin and COL1A1 increased at both the transcriptional and translational levels. It's only when anti-miR-135b was used that APC levels made sense. The other two proteins, although decreased, were still apparently not significantly different from siNC (which should have been compared [siNC vs. siMD1+a135b] by the authors). I think this suggests that LINCMD1 and miR-135b might be related albeit through unknown mediator(s). The authors must either refute this in a response or explain it in the Discussion.

Response: Thank you for your feedback, and we apologize for any confusion. In Figures 3A and 3B, we confirmed that LINCMD1 directly interacts with miR-135b. Subsequently, we demonstrated that miR-135b overexpression in myometrium spheroid cells led to decreased APC expression and increased expression of β-catenin and COL1A1 (Figures 3C and 3D). In Figure 4, we confirmed an inverse relationship between mRNA and protein expression levels of APC and total β-catenin/β-catenin activity/COL1A1 in leiomyomas. Additionally, based on the density analysis in Figure 5E, we have revised the results section to indicate that anti-miR-135b transfection following LINCMD1 knockdown partially reversed the protein levels of APC, non-phosphorylated β-catenin at Ser45, total β-catenin, and COL1A1. We greatly appreciate your valuable suggestions.

Reviewer 2 Report

Comments and Suggestions for Authors

The paper you shared addresses a field-specific gap by exploring the role of long non-coding RNA (lncRNA), specifically LINCMD1, in the pathogenesis of leiomyomas (uterine fibroids). While the involvement of LINCMD1 in skeletal muscle differentiation has been studied, its role in leiomyoma development was unknown prior to this research. This study is one of the first to show that LINCMD1 is significantly downregulated in leiomyoma tissues compared to matched myometrium tissues. This suggests that LINCMD1 plays a role in leiomyoma development, particularly through its interactions with miR-135b, which are critical for aberrant Wnt/β-Catenin signaling in these tumors.

Regarding the methodology in this article, some improvements and additional controls could enhance the robustness of the study:

1)    The study analyzes tissue samples from 34 patients, which is relatively small. Increasing the sample size could improve the statistical power and the generalizability of the findings.

2)    The siRNA-based knockdown of LINCMD1 in the spheroid cultures was significant, but confirming that the observed effects are specifically due to LINCMD1 knockdown and not off-target effects could be done using multiple siRNAs targeting different regions of LINCMD1.

Author Response

Reviewer #2

The paper you shared addresses a field-specific gap by exploring the role of long non-coding RNA (lncRNA), specifically LINCMD1, in the pathogenesis of leiomyomas (uterine fibroids). While the involvement of LINCMD1 in skeletal muscle differentiation has been studied, its role in leiomyoma development was unknown prior to this research. This study is one of the first to show that LINCMD1 is significantly downregulated in leiomyoma tissues compared to matched myometrium tissues. This suggests that LINCMD1 plays a role in leiomyoma development, particularly through its interactions with miR-135b, which are critical for aberrant Wnt/β-Catenin signaling in these tumors.

Regarding the methodology in this article, some improvements and additional controls could enhance the robustness of the study:

1)    The study analyzes tissue samples from 34 patients, which is relatively small. Increasing the sample size could improve the statistical power and the generalizability of the findings.

Response: Thanks for the suggestions. We included a paragraph on Statistics and Power Analysis in the M&M section.

Line 333-336: “Assuming a minimal detectable difference of 25% between leiomyoma and myometrium pairs, with an expected standard deviation of 25%, at least 16 paired samples are required to achieve a statistical power of 0.80 at a significance level of 0.05 using a two-sided t-test.”

2)    The siRNA-based knockdown of LINCMD1 in the spheroid cultures was significant, but confirming that the observed effects are specifically due to LINCMD1 knockdown and not off-target effects could be done using multiple siRNAs targeting different regions of LINCMD1.

Response: Thanks for the suggestions. To minimize the risk of off-target effects, we compared our results with cells transfected with a negative control siRNA (siNC), which consists of random nucleotides. We believe this approach reduces the likelihood of off-target effects caused by siLINCMD1, which is specifically designed to target LINCMD1.

Round 2

Reviewer 1 Report

Comments and Suggestions for Authors

The authors have adequately addressed my comments.